# Identifying and Estimating the Location of Sources of Industrial Pollution in the Sewage Network

**DOI:** 10.3390/s21103426

**Published:** 2021-05-14

**Authors:** Magdalena Paulina Buras, Fernando Solano Donado

**Affiliations:** 1Institute of Telecommunications, Faculty of Electronics and Information Technology, Warsaw University of Technology, ul. Nowowiejska 15/19, 00-665 Warsaw, Poland; magdalena.buras.stud@pw.edu.pl; 2Blue Technologies sp. z o.o., ul. Puławska 266/221, 02-684 Warsaw, Poland

**Keywords:** random forest, XGBoost, water pollution, machine learning, sensors

## Abstract

Harsh pollutants that are illegally disposed in the sewer network may spread beyond the sewer network—e.g., through leakages leading to groundwater reservoirs—and may also impair the correct operation of wastewater treatment plants. Consequently, such pollutants pose serious threats to water bodies, to the natural environment and, therefore, to all life. In this article, we focus on the problem of identifying a wastewater pollutant and localizing its source point in the wastewater network, given a time-series of wastewater measurements collected by sensors positioned across the sewer network. We provide a solution to the problem by solving two linked sub-problems. The first sub-problem concerns the detection and identification of the flowing pollutants in wastewater, i.e., assessing whether a given time-series corresponds to a contamination event and determining what the polluting substance caused it. This problem is solved using random forest classifiers. The second sub-problem relates to the estimation of the distance between the point of measurement and the pollutant source, when considering the outcome of substance identification sub-problem. The XGBoost algorithm is used to predict the distance from the source to the sensor. Both of the models are trained using simulated electrical conductivity and pH measurements of wastewater in sewers of a european city sub-catchment area. Our experiments show that: (a) resulting precision and recall values of the solution to the identification sub-problem can be both as high as 96%, and that (b) the median of the error that is obtained for the estimation of the source location sub-problem can be as low as 6.30 m.

## 1. Introduction

Discharges of industrial wastewater into the sewer network from industrial organizations require an official permit. These official permits include the maximum levels for a limited number of certain substances, restrictions on the level of pH and temperature, as well as the maximum allowed volume of oil and grease, for example. Radioactive material and persistent chemicals, among others, are explicitly mentioned and they must not exceed concentrations that are typical for domestic wastewater [1].

Unfortunately, not all industrial organizations carefully follow these limits, and they regularly dispose large amounts of harsh industrial waste illegally. As examples, discharges of sulfuric acid (H_2_SO_4_) to sewers could originate from applications, such as etching of semiconductors, accumulator acid, or the production of organic chemical substances [2]. Sodium hydroxide (NaOH) is widely used for cleaning of surfaces in metal processing in industrial applications [3], whereas discharges of sodium sulfate (Na_2_SO_4_) can be caused by the regeneration of cation exchange resins, which are used for softening of water in industrial water treatment [4]. Illegal discharges of such dangerous harsh industrial waste into sewage networks could be harmful in the biological stage of wastewater treatment plants (WWTP), its personnel, sewer pipes, and civilians.

Not all illegal discharges of harsh industrial waste can be detected at the WWTPs, due to wastewater dilution effects in sewer pipes through the network. Therefore, monitoring as close as possible to the point of discharge is necessary for avoiding any impairment of sewer operation and protecting neighbouring infrastructure and population against odours and explosive gas compositions.

### 1.1. Case Study—Baarle-Nassau, The Netherlands

In December 2016, the WWTP of Baarle-Nassau, The Netherlands (NL), failed [5]. The operator noticed that biological treatment stage failed completely, since the pH level in the aeration tank was extremely acidic; with a pH level of nearly 1. In cooperation with law enforcement agencies (LEA), research staff of the water board tried to backtrack the source of the low pH substance through the sewage system manually, by taking samples at certain distances through the pipe network.

Unfortunately, heavy rainfall flushed the acidic residuals of the waste away from the pipeline and it prevented the final localization of the source.

The failure of the WWTP lasted three days. Only stopping, cleaning, and restarting of the WWTP was estimated to cost between 80,000 and 100,000 EUR [6].

### 1.2. Background and Related Work

In recent years, several sensor prototypes for monitoring wastewater composition at points that are further from the WWTP [7,8,9,10,11,12,13,14,15,16,17,18,19,20,21,22,23] have been proposed and studied. These sensors (electrochemical sensors, optical sensors, mass spectrometry, ion spectrometry, etc.) can be mounted within manholes and main sewer lines, and aim at detecting the presence or concentration of certain pollutants.

In this paper, we consider that pollution of wastewater is a rare event and, therefore, we assume that there is only a single polluting source at a time in the monitored sewer network.

Figure 1 shows a wastewater network that corresponds to a sub-catchment area involving 42 buildings of an European city. A polluting source, $s_{i}$, may inject a limited amount of a pollutant into the sewage system in a short amount of time. Because of the dispersion effects in hydraulic channels, the injected pollutant can be detected in smaller concentrations in nearby pipes transporting wastewater in the direction of the WWTP. In order to illustrate this effect, consider that 100 L of sulfuric acid are discharged at building B13 of Figure 1 at 03 h 00 min. Figure 2 shows time-series of pH and electrical conductivity (EC) values measured at different manholes along the path from its source (building B13 connected to manhole 09) to the closest point to the WWTP (manhole 16).

Because of the dilution effects and limited sensitivity of the sensor devices, the pollutant can only be detected in those pipes where the diluted concentration exceeds the minimum limit of detection of the sensor. Given that the flows of wastewater are acyclic in a sewage network, the set of pipes where the pollution from a particular source can be detected form a directed acyclic sub-graph, namely $G(s_{i})$, of the sewage network. It shall be noted that, given two potential polluting source $s_{i}$ and $s_{j}$, the corresponding sub-graphs, $G(s_{i})$ and $G(s_{j})$, where the pollutant that can be detected may have sewer pipes or edges in common. A sensor device placed in such edges is not able to discern with a single measurement whether the detected pollutant originates from either $s_{i}$ or $s_{j}$. Therefore, most of the work that was carried out in the identification of source of pollution in water distribution systems, drainage networks, and wastewater networks related to the idea of finding a match between an input time-series of sensor measurements and a modelled (or simulated) time-series of concentration or physical parameter values for different sources. A summary of these approaches is briefly presented below.

Di Cristo et al. [24] provide a mathematical programming formulation for finding the best match of a given input time-series of measurements to one simulated time-series of pollutant concentration values for different sources. We proceed to briefly explain their formulation. Let $C_{k}^{t}$ be the concentration that is measured at time *t* by a sensor node located at point *k* in the network. Let $\hat{C_{k}^{t}}\left(j,S_{j}^{t}\right)$ be the simulated contaminant concentration at time *t* at point *k* in the network, which is an implicit nonlinear function of its source location *j* and of the input concentration magnitude $S_{j}^{t}$. The values of $\hat{C_{k}^{t}}\left(j,S_{j}^{t}\right)$ are usually calculated through simulations that are based on modelling of the channel hydraulics. Subsequently, Di Cristo et al. [24] propose the following fitness function $f_{j}^{t}$ for every time *t* and location *j*:(1)$$f_{j}^{t}=\sum_{k=1}^{M}{\left(\hat{C_{k}^{t}}\left(j,S_{j}^{t}\right)-C_{k}^{t}\right)}^{2}$$

A fitness function is a particular type of objective function that is used to summarise, as a single figure of merit, how close a given design solution is to achieving the set aims. Di Cristo et al. [24] consider that the source of the pollution is the location *k* with the maximum average of the fitness function over all time-instants *t*. Other authors propose several other similar optimization functions in [9,25,26]. In our opinion, the performance of these approaches depends on the time alignment of the values between the measured and modelled time-series.

Other approaches involve the usage of machine learning algorithms. Jalal et al. [27] analyse the efficiency of two classification algorithms— decision trees and support-vector machines (SVM)—trained on data obtained from water treatment station (WTS) in Tunisia. For their study, Jalal et al. [27] employ a wireless sensor network (WSN), where the sensors measure inter-alia, pH, temperature, concentration of arsenic, magnesium, and calcium. The solution based on decision trees resulted in 82% of precision and 83% of recall. For the SVM-based solution, these metrics reached 84% and 85%, respectively.

Macas et al. [28] propose the usage of a machine learning algorithm for detecting anomalies in WTSs. Because of the insufficient amount of labelled data, Macas et al. [28] propose an unsupervised algorithm. Macas et al. [28] reject the usage of k-means algorithm and SVM, since they do not detect temporal patterns in the multidimensional data, thus becoming prone to false positive errors. Instead, the researchers opt for classical recurrent networks, like long-short term memory (LSTM) networks. Nevertheless, the idea was abandoned, as these structures were not producing satisfying performances with noisy data. Eventually, the researchers proposed a spatio-temporal autoencoder for anomaly detection, STAE-AD. In a nutshell, their model extracts the most important features allowing to detect the main trend in time and space from the multidimensional data using an attention-based convolutional LSTM. Based on these features, a convolutional decoder tries to recreate a statistical correlation matrix. The less similar thus obtained values are compared to the real ones. This model was trained using approximately 1,000,000 records, consisting of usual and contaminated data. The values of precision and recall of that model reached 96% and 82%, respectively.

Chachula et al. [29] proposed a data fusion algorithm for solving the pollution source localization problem in wastewater networks. The approach does not utilize machine learning algorithms. Instead, it follows the general data fusion model of Mitchell [30]. Data fusion parameters, such as water speed, dispersion of each substance, substance quantification, etc., have to be carefully chosen so as to match real hydraulic conditions.

In this article, we provide a solution to the problem of detecting and identifying a wastewater pollutant, as well as estimating the distance between the source point where the pollutant was injected and its point of detection, given a time-series of observations of a monitoring device with multiple sensors. The solution approach that is presented in this article considers machine learning algorithms. As a case study, we consider that the monitoring devices are equipped with pH and EC sensors (such as those used by the Micromole device (Blue Technologies sp. z o.o., Warsaw, Poland) [7,8]), and that we are interested in discerning between the three major industrial pollutants of urban wastewater mentioned above.

## 2. Materials and Methods

In this article, we divide the problem into two smaller sub-problems and handle them separately in order to increase the solution’s accuracy. These two problems are *pollutant detection and identification* and *pollutant source location estimation*.

In this article, we consider the case when labelled data are available. Labelled data can be obtained empirically—by performing discharges in a real sewage network and recording measurements in other points—or through computer simulations of discharge events using specialized software. For such cases, supervised machine learning algorithms may be considered. In this article, we evaluate a classification algorithm for the detection and identification of a pollutant, and a regression algorithm for the estimation of the pollutant’s source location.

If the classification algorithm can detect and identify a pollutant as one of the toxic substances, then the regression algorithm is used for the estimation of the pollutant’s source location. If the classification algorithm is unable to detect a pollutant in the given input data set, the time-series is considered to be ‘clean‘, *viz*. not contaminated by a pollutant for the purposes of this work, and no more analysis is performed.

### 2.1. Pollutant Detection and Identification

In Figure 3, we show measurement values of pH and EC of clean measurements of wastewater. These values do not display significant and rapid changes in values of neither pH nor EC. However, the values may oscillate through the day, as the usage of water (and, consequently, discharged wastewater) changes according to the daily rhythm of the city inhabitants.

The task of the detection and identification model is to determine whether the examined sensor samples are contaminated and if so, identify which of the known polluting substances is mixed with the wastewater, while considering the normal fluctuations in wastewater parameters that may occur during the day. The toxic substances considered in this article are: sulfuric acid, sodium sulfate, and sodium hydroxide, as previously discussed in Section 1. Table 1 presents the characteristics of the three substances.

When discharged into the sewage network, each one of these substances is accountable for different changes in pH and EC values in wastewater.

For the measurements that are contaminated with sulfuric acid, values of pH significantly drop at the moment of the discharge. Moreover, values of EC increase rapidly around the same time, as observed in Figure 4.

Pollution with sodium hydroxide results in a significant increase of pH values and in a less intense decrease in measurements of EC, as illustrated in Figure 5.

Contamination with the third and the last substance, sodium sulfate, is characterized by a significant decrease in EC values and a slight oscillation of pH values, as shown in Figure 6.

In order to solve the detection and identification sub-problem, we use a random forest classifier. Random forest is a supervised machine learning algorithm that can be used to solve the classification and regression problems [31]. A random forest is built from many decision trees that merge their predictions together to increase the final model’s accuracy—this technique is called ensemble learning [32]. Despite consisting of decision trees, that can be called weak learners, random forest manages to perform well in many problems. Every tree gets to make its prediction based on a different set of the features describing the examined sample—this is called feature randomness. In classification problems, the final answer of the model is the dominant of all the choices. An interesting example demonstrating how powerful this algorithm can be is described in [33]. The researchers used the random forest classifier to optimise locations for the safety cameras by the busy roads in Morocco in order to prevent car accidents. They trained, tested, and compared several classification methods, including SVM, K-nearest neighbours, K-means, gaussian naive bayes, and linear discrimination analysis algorithms. Random forest classifier proved to be the most effective with the accuracy of 95% that was measured for the cross-validation set.

One of the possible solutions to our detection and identification sub-problem could be achieved by treating all of the measured values of EC and pH as the input for the model. Nevertheless, such a model would have the following shortcomings: (a) a large amount of time to train and compute final predictions, (b) prone to classification errors, if some of the sensor measurements get lost or duplicated, and (c) the lists of examined measurements could be of a different size than expected.

When considering the previously mentioned shortcomings, we opt for describing time-series of pH or EC values as a selected set of features that enable the discrimination of substances by a classifier. The features proposed in this article for this sub-problem are: mean value, standard deviation, minimum value and maximum value for pH and EC samples, and id of the sensor that took the measurements. Because a single sensor monitors a given location, the sensor id is used for identifying the point of measurement.

### 2.2. Pollutant Source Location Estimation

The second sub-problem concerns the estimation of the distance from the measurement point (known by the sensor id) and the discharge source point in the sewage network.

When considering the differences in the time-series that are depicted in Figure 2 between different points of measurement, we can observe that, the farther the sensors are, the less significant the changes are for the pH and EC values caused by the discharge: these changes diminish as the distance increases and flows from other sources aggregate. This phenomenon is explained by the dilution, advection, and dispersion hydraulic process in a channel.

For sulfuric acid, the minimum pH and maximal EC values provide an indication of a change in the distance. As the distance increases, these values tend to be much higher and lower, respectively. Figure 7a,b use the same scale to emphasise these changes.

The samples contaminated by sodium sulfate that were collected at different distances from its source differ in their maximum EC value: the higher the distance, the lower the maximum value. The values of pH oscillate slightly, as observed in Figure 8a,b.

The samples contaminated with sodium hydroxide collected at different distances from its source differ mostly in their maximum pH value: the higher the distance, the lower the maximum value. Values of EC do not change significantly, as observed in Figure 9a,b.

The following features are used to describe properly the nature of the volatility of the distance: substance name, sensor ID, mean value, standard deviation, minimum, maximum, first quartile, second quartile, the third quartile for pH and EC, and the hour of the measurement. Quarterlies are used to ensure more detailed description of the pH and conductivity changes. Substance name is added for pollutants differentiation. We include the hour as one of the input features for compensating against the normal daily fluctuations in pH and EC values in wastewater.

For solving this subtask, the XGBoost algorithm is used. XGBoost, namely extreme gradient boosting, is an improved version of the gradient boosting algorithm in terms of efficiency and scalability. Like random forest, this algorithm belongs to the ensemble methods and uses decision trees as weak learners.

The provided model estimates the distance from the pollution source. Given that it can only be located along the sewage network and the wastewater in the network cannot flow upstream, we can determine the source as one of the antecedent manholes located closest to the predicted distance from the sensor that took the measurements. In the case of more than one probable manholes, the model’s predictions for surrounding sensors are taken into consideration.

## 3. Results

### 3.1. Data and Experiments

Figure 1 shows the structure of the simulated network. The network consists of 42 buildings and 15 manholes, where manhole number 16 is the sink of the depicted sub-catchment area.

All of the flow and discharge simulations were performed using the software package ++SYSTEM Isar [34], where the capabilities were extended by a reaction and transport model based on the concept of total alkalinity.

Table 2 presents starting and ending manhole, the flow rate, water level, and velocity for all sewers along the flow path for a discharge in a building connected to manhole number 09 of the network in order to describe the hydraulic situation in this network immediately before the discharge event. The flow of wastewater was calculated based on the number of inhabitants for all buildings connected to each sewer and their freshwater consumption. The latter shows a typical daily pattern, as can be seen in Figure 10. Using the flow of wastewater at 03 h 00 min of a day and the flow at 08 h 00 min, we defined a low and a high flow scenario. This allows us to take a best-case and worst-case view of transport and dilution processes.

We simulated the dilution behaviour of the sewage network that is shown in Figure 1 over time when industrial wastewater is discharged from each building separately. A simulated event consists in the discharge of 100 L of a substance from Table 1 in a selected building at either low (03 h 00 min) or high (08 h 00 min) wastewater flow conditions. Since there are 42 potential sources in our network, there are 252 different simulation scenarios, when considering the different (2) flow conditions and (3) substances.

From each simulated event, we obtained a set of time series of EC and pH measurements, as observed through the traversing manholes in the network. An experiment is defined as a time-series of measurements of EC and pH at a single manhole from a given simulation. Since there are 285 combinations of buildings and manholes, due to the flow directions of the wastewater network, the total number of experiments was 1710, when considering the different (2) flow conditions and (3) substances.

The features that are mentioned in Section 2.1 and Section 2.2 were extracted for each one of the 1710 experiments.

The computed features summarize the most important information regarding the experiments for a single simulated day of observations in the sewage network for different discharge events. Based on these extracted features for this single day, we generated synthetic feature data in order to include more data corresponding to days where the hydraulic conditions in the sewage network are similar. In other words, a data generator was implemented for providing synthetic extracted features, instead of synthetic measurements from sensors. The data generator takes the original feature values and multiplies them by a noise factor, resulting in synthetic features in the range of 0.95 and 1.05 of the original feature value.

When considering the synthetic features, the total number of dataset features considered in this section was 4909 records. Approximately 67% of them was used as a training set and the remaining 33% as the test set.

The training and testing algorithms—including the feature extraction and data generator methods—were implemented in Python. The random forest classifier was imported from the sci-kit learn machine learning library [35]. The XGBoost algorithm was imported from XGBoost Python package [36].

### 3.2. Feature Analysis

With the random forest classifier, it is possible to check feature importance of the training data, namely: what is the impact of every single feature on the model’s final decision. Features important for that model are presented more in detail in Table 3.

As we can observe from Table 3, features with the highest impact on the model’s final predictions are the values of standard deviation for EC and pH values, mean, and minimum values of pH. Together, these four features account for almost 90% of the final decision. Standard deviation is a metric that describes how close to the mean value are other samples. The mean value itself is much less important to the model. Interesting, the fourth most important feature is a minimum pH value −6.38%.

Concerning the pollutant source location estimation problem, as we can observe from Table 4, the standard deviation of EC and pH were found to be the most useful by the model, as well as their maximum values. Surprisingly, the substance name and hour of the measurement observation had very little impact on model’s final prediction.

### 3.3. Pollutant Detection and Identification

Precision and recall metrics were computed to ensure that model’s performance is monitored in an informative way.

In Table 5, we present such values for the test set. We can see that all of the metrics values are higher that 93%. The mean values of precision, recall, and F1 score reached 96%.

Because the location estimation model is used only once a contamination event is confirmed, from that point of view, mistaking the pollutants between each other is less harmful than misclassifying a contaminated measurement as a clean one, as the source detection model is not activated in that case. The very high value of the recall metric for the clean measurements proves that such a mistake would be very rare.

Figure 11 shows confusion matrix, which presents more detailed prediction error tendencies of the classifier. We can observe that, if model is misclassifying sodium sulfate measurements, then it is probably because it is confusing them with the clean probes. Moreover, if sulfuric acid measurements gets misclassified, it is because the model is confusing it with sodium sulfate or clean values. These errors are quite rare, as the classifier’s performance is high.

### 3.4. Pollutant Source Location Estimation

Table 6 gathers more detailed information on regression model’s performance. These results are obtained with the training set.

As we can see, for 25% of the samples, the error is 2.56 m and less. Error median is 6.30 m, which means that half of the mistakes made by the model are 6.30 m or lower.

Figure 12 shows how the error value changes as the sensor distance from the discharge point varies. The aim of this experiment was to check if the value of the error increases as the distance grows.

The lowest error values, as indicated by mean and the median, are noted for the distances between 90 and 120 m and they are 6.16 and 4.32 m, respectively. The highest error values are noted for distance values between 210 and 240 m: 14.20 and 10.37 m, respectively. The probable cause of these is the high inflow of wastewater into the utility network, which causes the pollutant to disband. This figure shows that the error values vary as the distance from the source increases, yet we cannot say that the prediction error is correlated with the distance value.

Finally, we analyse whether the model has the tendency to predict too high or too low values. The model is more likely to return too high predictions, as shown in Figure 13. The difference in these probabilities is two percentage points.

## 4. Discussion

When considering the protection of the WWTP, its infrastructure, network, and employees, it is important to avoid overlooking polluting discharge events. Overlooking pollution events yield situations where no mitigation action is carried out promptly for the WWTP protection. Overlooking pollution events relates to the number of false-negative results of a pollution detection algorithm. From our results, we observe that such situations may happen in 11% and 5% of the cases when sodium sulfate or sulfuric acid, respectively, are illegally disposed (see Figure 11). There is no such risk for sodium hydroxide discharges.

Once a harmful discharge is detected, WWTP personnel may apply specific mitigation strategies for the protection of the WWTP infrastructure against a specific threat. Correct identification of the pollutant is of importance in this second step. In less than 9% of the cases, the proposed algorithm is not able to discriminate between sodium sulfate and sulfuric acid (see Figure 11). We suspect such situations occur at large distances from the source, as the measured value of pH cannot be distinguished from clean wastewater, but only detected by a significant increase in EC. However, in such cases, it is most likely that the applied mitigation and protection measures to the WWTP are similar, since the pH values of the polluted wastewater at the WWTP would be almost normal.

The estimation of the location of the source of pollution is important for criminal investigations that were carried out by LEAs, as prevention mean further harmful discharges in the future. The algorithm proposed in this article would allow criminal investigators to narrow down the suspected area of the discharge source by a given radius with a span of 14.20 m upstream from the point of measurement, in the worst case, in a distance of 210 m. Given the distances between industrial lots, we consider that the provided accuracy is sufficient for criminal investigation purposes. However, our method is not capable of providing the exact localization of the source, but a region upstream from the measurement point that is delimited by two arcs in the given catchment area. Further research should be carried out in this direction.

## 5. Conclusions

In this article, we provided a machine learning solution for the problem of identifying a pollutant in wastewater and determining its point of origin. The algorithms were tested considering a real sewage network from Europe and simulated discharged experiments of three different substances.

In brief, the results of the substance identification problem are very accurate, with 96% of precision and recall on average. Concerning the estimation of the location of the source, errors up to 14.2 m in a range of 210 m were observed in the worst case. Such errors are tolerable for the purpose of initiating a manual checking procedure of the point of origin of the pollutant source by environmental criminal investigators. However, it should be clarified that our method is not capable of providing the exact localization of the source, but a region that is delimited by two arcs upstream from the measurement point within a catchment area. The proposed method will be extended to identify a set of potential inflow points where the source may be located.

In addition, it will be further evaluated with larger sewage networks, more substances, and different sewage hydraulic conditions in further publications. A benchmark of different localization methods–such those described in [24,27,28,29,30]—for an estimation of the pollution source could be considered as an important step forward in this area, as well as adapting effective existing methods for detection of polluting sources in similar environments, such as in [37,38].

## Figures and Tables

**Figure 1 sensors-21-03426-f001:**
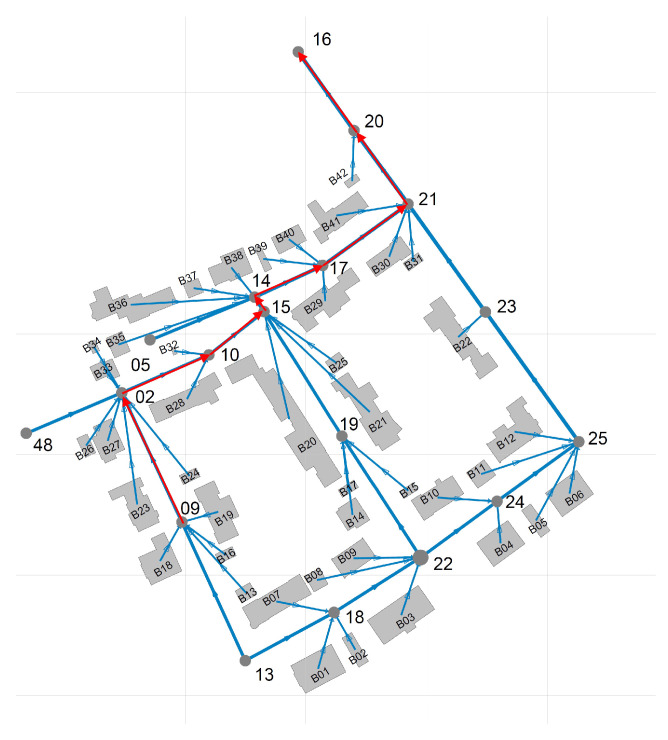
Sewer network of undisclosed sub-catchment area of an European city. Red arrows indicate the flow path starting from manhole No. 09.

**Figure 2 sensors-21-03426-f002:**
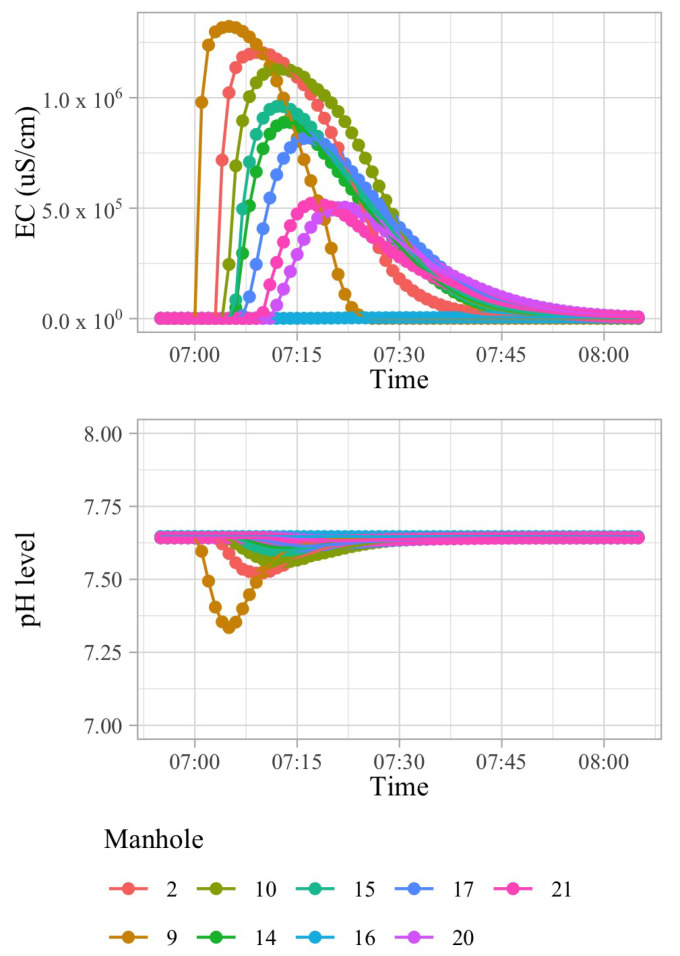
Electrical conductivity and pH peak broadening and flattening caused by dispersion, as seen by different manholes, when one hundred liters of sulfuric acid are discharged at location B13 of Figure 1.

**Figure 3 sensors-21-03426-f003:**
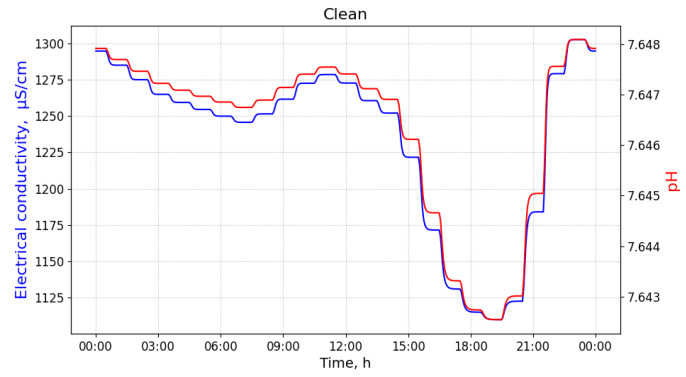
Measurements of pH and electrical conductivity for the clean utility network content. Values of the horizontal axis represent time of the day. Measured at the manhole 48.

**Figure 4 sensors-21-03426-f004:**
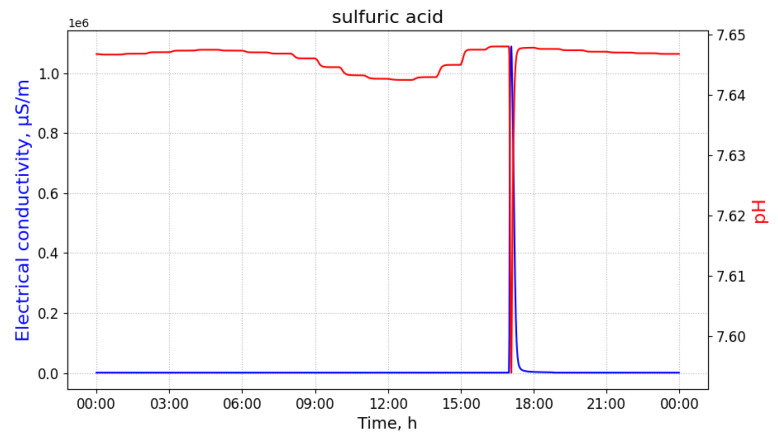
Measurements of pH and electrical conductivity for the utility network content contaminated with sulfuric acid. The values of the horizontal axis represent time of the day. Measured at the manhole 02, discharged at the building B33 with 100 L of the pollutant at 8 a.m.

**Figure 5 sensors-21-03426-f005:**
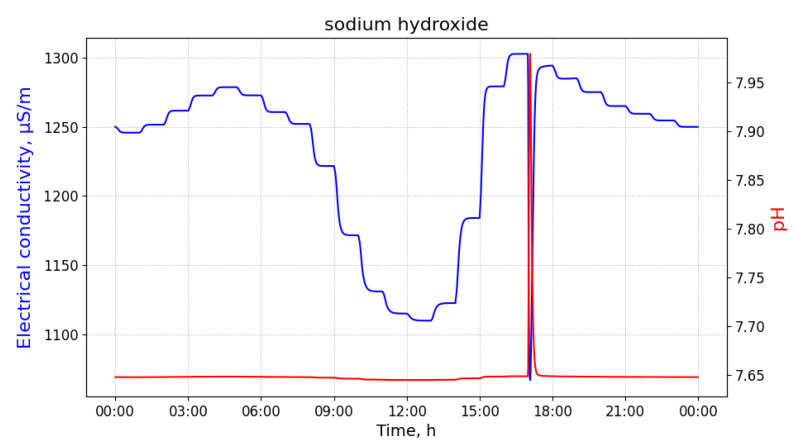
The measurements of pH and electrical conductivity for the utility network content contaminated with sodium hydroxide. Values of the horizontal axis represent time of the day. Measured at the manhole 02, discharged at the building B33 with 100 L of the pollutant at 8 a.m.

**Figure 6 sensors-21-03426-f006:**
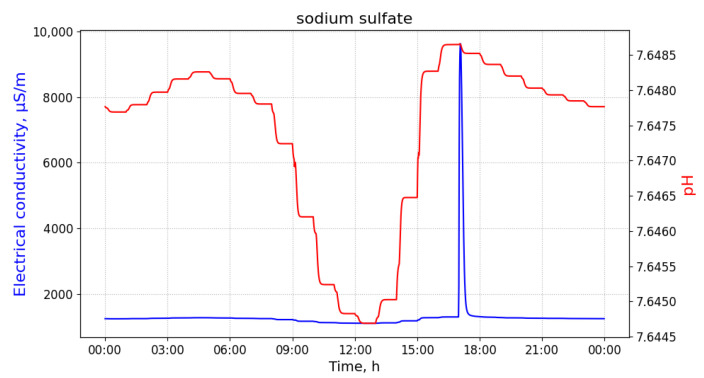
The measurements of pH and electrical conductivity for the utility network content contaminated with sodium sulfate. Values of the horizontal axis represent time of the day. Measured at the manhole 02, discharged at the building B33 with 100 L of the pollutant at 8 a.m.

**Figure 7 sensors-21-03426-f007:**
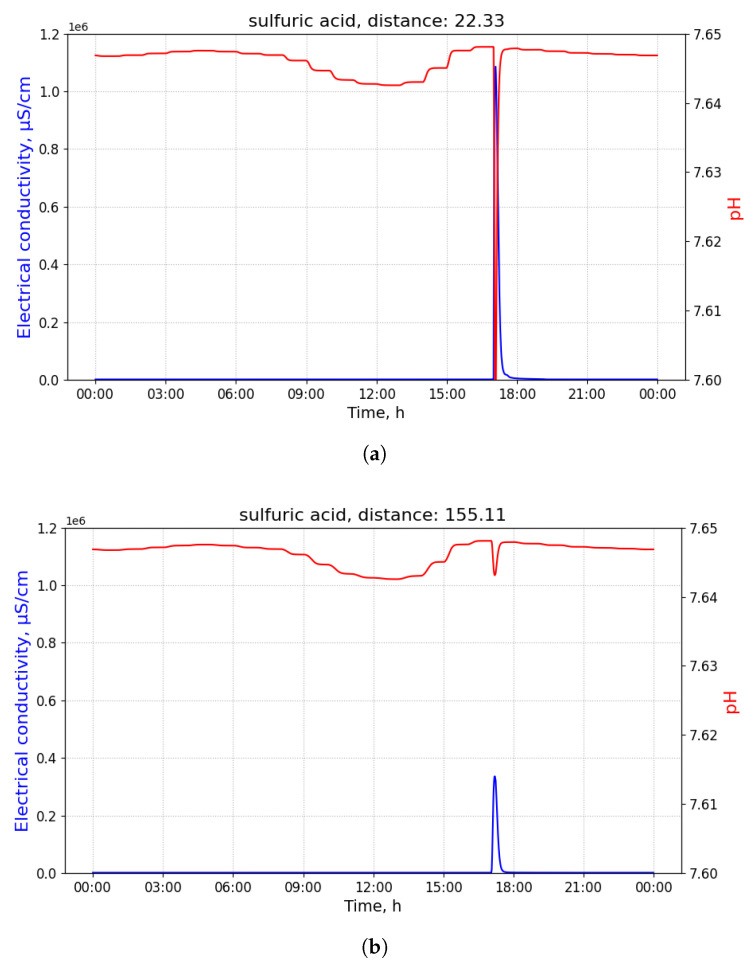
The measurements of pH and electrical conductivity (EC) for the utility network content contaminated with sulfuric acid conducted at different distances. Values of the horizontal axis represent time of the day. (**a**) Measurements of pH and EC for the utility network content contaminated with sulfuric acid conducted at the distance of 22.33 m from the contamination source. Measured at the 02 manhole, discharged at the building B34 with 100 L of the pollutant at 8 am. (**b**) Measurements of pH and EC for the utility network content contaminated with sulfuric acid conducted at the distance of 155.11 m from the contamination source. Measured at the 20 manhole, discharged at the building B32 with 100 L of the pollutant at 8 a.m.

**Figure 8 sensors-21-03426-f008:**
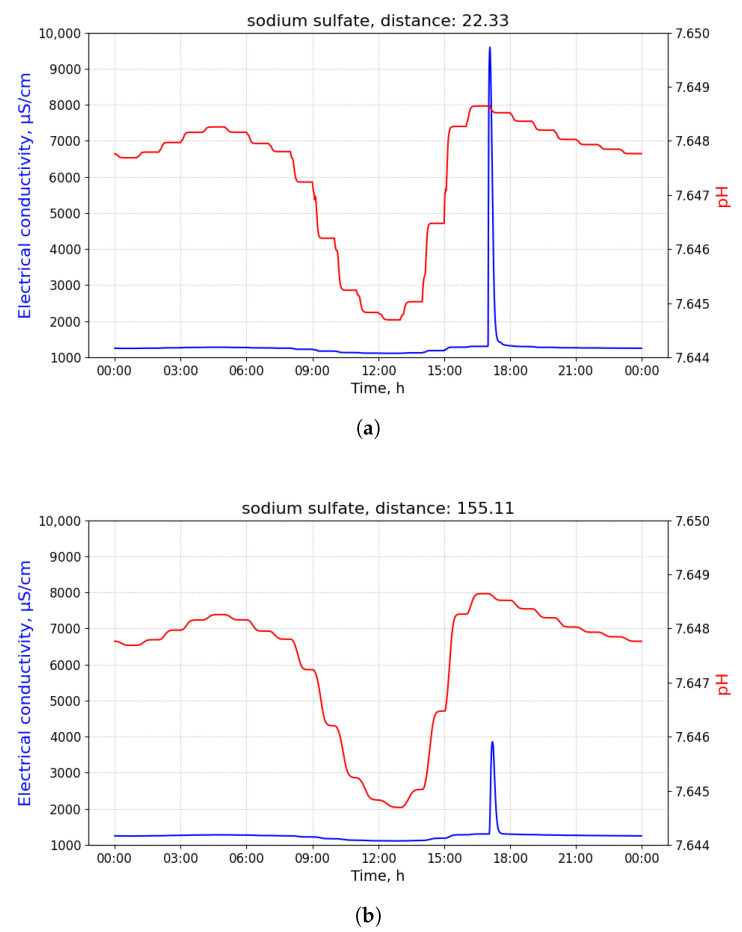
Measurements of pH and electrical conductivity (EC) for the utility network content contaminated with sodium sulfate conducted at different distances. Values of the horizontal axis represent time of the day. (**a**) Measurements of pH and EC for the utility network content contaminated with sodium sulfate conducted at the distance of 22.33 m from the contamination source. Measured at the 02 manhole, discharged at the building B34 with 100 L of the pollutant at 8 am. (**b**) Measurements of pH and EC for the utility network content contaminated with sodium sulfate conducted at the distance of 155.11 m from the contamination source. Measured at the 20 manhole, discharged at the building B32 with 100 L of the pollutant at 8 a.m.

**Figure 9 sensors-21-03426-f009:**
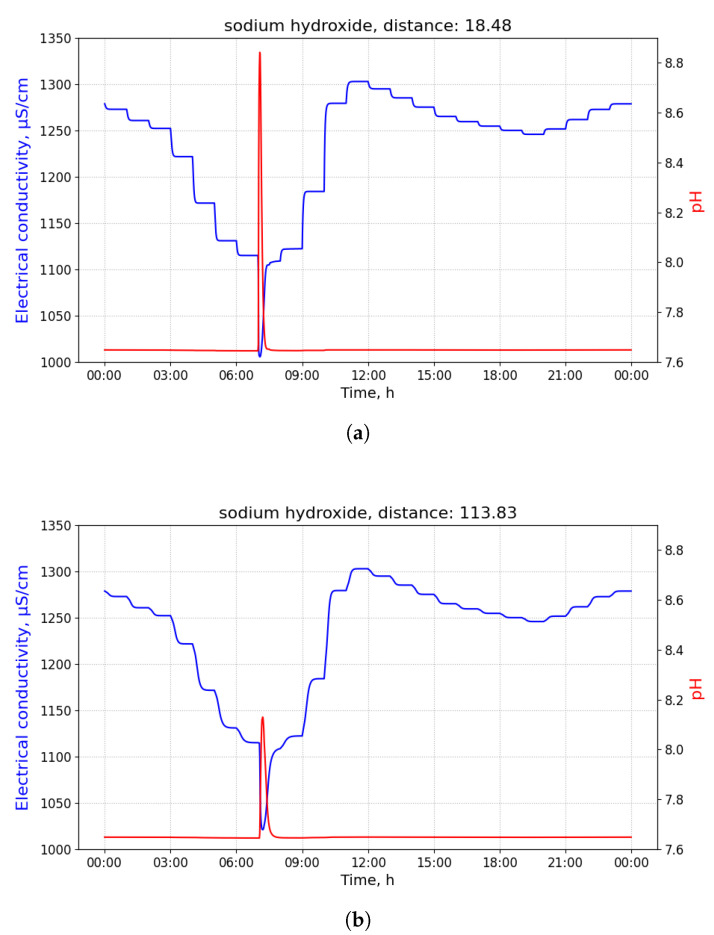
Measurements of pH and electrical conductivity (EC) for the utility network content contaminated with sodium hydroxide conducted at different distances. Values of the horizontal axis represent time of the day. (**a**) Measurements of pH and EC for the utility network content contaminated with sodium hydroxide conducted at the distance of 18.48 m from the contamination source. Measured at the 09 manhole, discharged at the building B18 with 100 L of the pollutant at 3 a.m. (**b**) Measurements of pH and EC for the utility network content contaminated with sodium hydroxide conducted at the distance of 113.83 m from the contamination source. Being measured at the 10 manhole, discharged at the building B18 with 100 L of the pollutant at 3 a.m.

**Figure 10 sensors-21-03426-f010:**
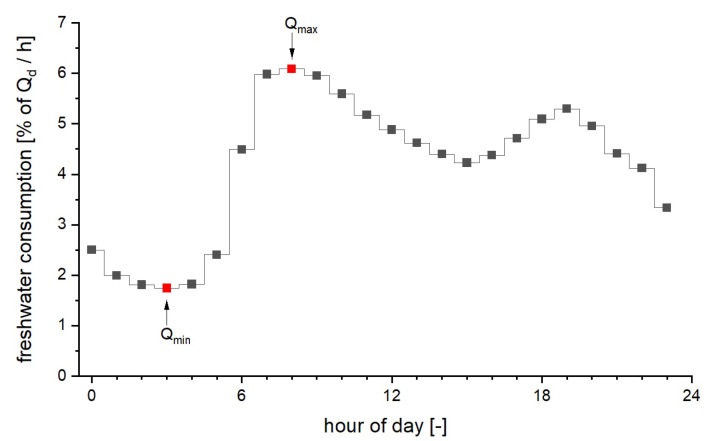
Diurnal pattern of freshwater consumption.

**Figure 11 sensors-21-03426-f011:**
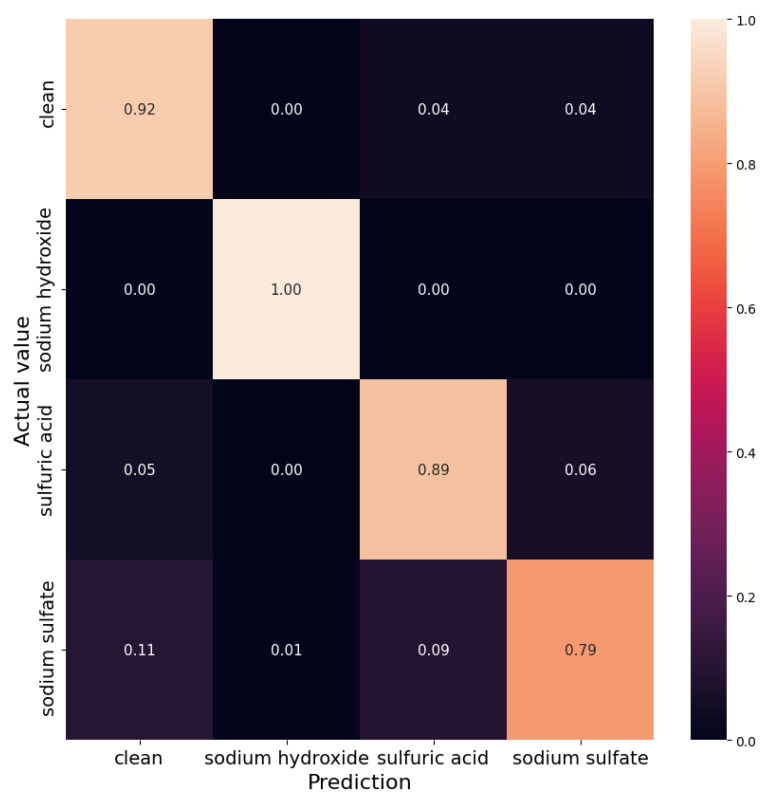
Confusion matrix for the pollutant classification based on the pH and electrical conductivity measurements.

**Figure 12 sensors-21-03426-f012:**
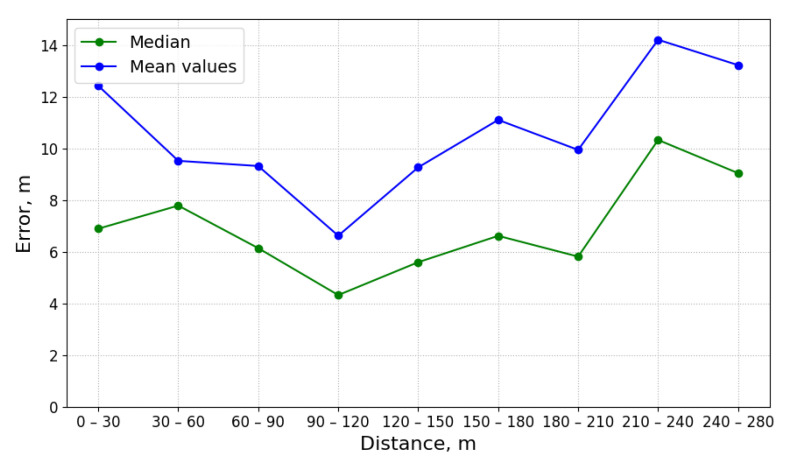
This figure presents the correlation between the prediction error, mean value and the median, and the distance at which the measurements were taken.

**Figure 13 sensors-21-03426-f013:**
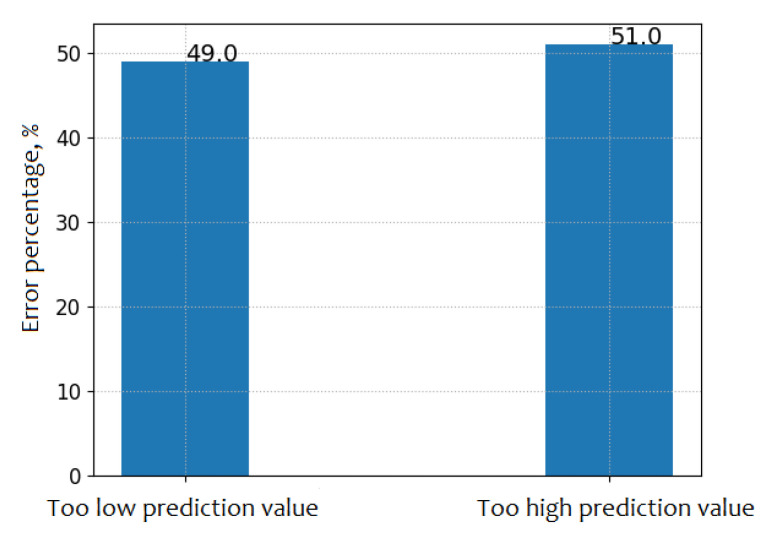
This figure presents the percentages of the too low and too high values predicted by the model.

**Table 1 sensors-21-03426-t001:** Substances used in simulations.

Short	Substance Name	pH	EC, mS/cm
H_2_SO_4_	sulfuric acid	1	1400
NaOH	sodium hydroxide	12	1
Na_2_SO_4_	sodium sulfate	9	12

**Table 2 sensors-21-03426-t002:** Hydraulic characteristics of the modelled sewer network.

Sewer	From	To	Length (m)	Water Level (cm)	Flow Rate (L/s)	Velocity (m/s)
1	9	2	58	0.4	0.02	0.09
2	2	10	38	0.4	0.03	0.09
3	10	15	27	0.4	0.03	0.09
4	15	14	6	0.5	0.05	0.13
5	14	17	29	0.5	0.07	0.13
6	17	21	42	0.5	0.08	0.13
7	21	20	36	0.8	0.16	0.16
8	20	16	38	1.2	0.16	0.20

**Table 3 sensors-21-03426-t003:** Feature importance—pollutant detection and identification.

Feature	Importance
Mean pH value	6.39%
Standard deviation of pH	35.83%
Maximum pH value	1.46%
Minimum pH value	6.38%
Mean conductivity value	3.82%
Standard deviation of conductivity	38.00%
Maximum conductivity value	2.30%
Minimum conductivity value	2.30%
Sensor ID	3.51%

**Table 4 sensors-21-03426-t004:** Feature importance—pollutant source location estimation.

Feature	Importance
Mean pH value	2.04%
Standard deviation of pH	21.01%
Maximum pH value	7.08%
Minimum pH value	4.27%
First quartile of pH	2.51%
Second quartile of pH	2.61%
Third quartile of pH	2.79%
Mean conductivity value	3.42%
Standard deviation of conductivity	24.16%
Maximum conductivity value	8.40%
Minimum conductivity value	5.27%
First quartile of conductivity	3.09%
Second quartile of conductivity	1.46%
Third quartile of conductivity	2.41%
Sensor ID	2.44%
Measurement hour	3.21%
Substance name	3.84%

**Table 5 sensors-21-03426-t005:** Classification accuracy—pollutant detection and identification.

Class	Precision	Recall	F1 Score	Number of Records
Clean	94%	97%	96%	352
Sulfuric acid	98%	96%	97%	421
Sodium sulfate	98%	96%	97%	432
Sodium hydroxide	94%	97%	95%	415
Sum/Mean	96%	96%	96%	1620

**Table 6 sensors-21-03426-t006:** Prediction accuracy—pollutant source location estimation.

Metrics	Error Value
First quartile	2.56 m
Second quartile (median)	6.30 m
Third quartile	14.25 m
Mean value	10.13 m

## Data Availability

Restrictions apply to the availability of these data. Data was obtained from Steffen Krause and Christoph Wöllgens of the Universität der Bundeswehr München and are available from the authors with the permission of Universität der Bundeswehr München.

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
