# Peer review of "Identifying and Estimating the Location of Sources of Industrial Pollution in the Sewage Network"

_sensors, 2021, doi:10.3390/s21103426_

Round 1

Reviewer 1 Report

see the file attached

Reviewer 2 Report

Comments to sensors 1170817

Sudden or unexpected industrial wastewater discharge into urban sewer network, poses a big threat to the normal operation of municipal wastewater treatment plant, and even leads to unattainable WWTP effluent water quality. Location of potential industrial wastewater discharge is a challenging task. This manuscript aims to develop a method for identifying and estimating the location of sources of industrial pollution in the sewage network based on on-line pH and electric conductivity data. From this perspective, a machine-learning method of XGBoost was employed.

After review this paper, my impression is this manuscript is less convincing, and no specific methods and discussion were provided to support your statement for industrial pollution source location. Therefore, I didn’t recommend publication for this manuscript.

Below are my major concerns for this manuscript:

  1. Abstract: This part should be re-written. It mainly introduces the background of SYSTEM project, instead of purpose, methods and findings of this study.
  2. Introduction: No literatures review was presented. I noticed that in Section 2.1, you listed some related work. However, this is still not enough for this field, e.g., detection of urban drainage network using inverse problem method.
  3. The appearance of Fig.1(a) seems strange, You stated the role of dispersion in flattening the concentration peak. I didn’t find the relationship of this discussion with industrial pollution source location.
  4. Location of industrial pollution source is an inverse problem using on-line data. Your manuscript didn’t provide any information for this method. You mentioned XGBoost algorithm and then used it to present feature importance - location estimation. However, we don’t know how to locate the pollution source.

Generally, no clear methods and findings were put forward for estimation and location of industrial pollution source discharge.

Reviewer 3 Report

The paper could be interesting, but some flaws need to be amended.

Introduction: lines 68-73 are unuseful

Section 2.1 "Related Works" could be a part of "introduction"

The sewage net should be characterized in terms of waste water volume, speed along the branches, crossing times, distances among manholes...

Fig. 1 should be redrawn to be better readable and described in the text 

The methods (e.g. random forest classifier) must be clearly presented and the experiment should be better detailed, also including a description of the sensors and of the data acquisition and transmission system

Fig. 2 and following ones: the "time" in the x-axys could be misleading: please clarify wether it represents a time of day or elapsed time

The description of the experiment is very poor.

The "discussion" is lacking.

The acronyms should be explained at their first use.

Some attention to English should be paid

Round 2

Reviewer 1 Report

Congratulations to the authors for their very detailed revisions, which I think will be accepted for publication after some careful proofreading of the article

Author Response

Dear Reviewer,

Thank you very much for your support and comments for this publication. We revised the usage of the language and some corrections were made accordingly.

All the best,

Dr. Fernando Solano

Reviewer 3 Report

The major flaws were amended.

Some small issues remain to be solved:

line 102: "fitness"?

line 213-214: please verify, there is something missing

line 262: please rewrite

line 280: further? better farther

line 313: "predicts"? better "estimates"

line 421-424: please rewrite

line 451-452: "...an area of a catchment area..." ?

line 463: corroboration ?

Please verify that all the Acronyms are properly cited in the text at their first use and, if you want to mantain the final Abbreviations table, verify to include all and only the cited ones (e.g. IoT is never used...) 

A final proof-reading is recommended.

Author Response

Dear Reviewer,

Thank you very much for your support and comments for this publication. We revised the usage of the language - following all your suggestions - and corrections were made accordingly.

All the best,

Dr. Fernando Solano